# Antitumor Effect of Açaí (*Euterpe oleracea* Mart.) Seed Extract in LNCaP Cells and in the Solid Ehrlich Carcinoma Model

**DOI:** 10.3390/cancers15092544

**Published:** 2023-04-28

**Authors:** Walbert Edson Muniz Filho, Fernando Almeida-Souza, André Alvares Marques Vale, Elis Cabral Victor, Mirtes Castelo Branco Rocha, Gabriel Xavier Silva, Amanda Mara Teles, Flavia Raquel Fernandes Nascimento, Carla Junqueira Moragas-Tellis, Maria do Socorro dos Santos Chagas, Maria Dutra Behrens, Daiana de Jesus Hardoim, Noemi Nosomi Taniwaki, Josélia Alencar Lima, Ana Lucia Abreu-Silva, Rui M. Gil da Costa, Kátia da Silva Calabrese, Ana Paula Silva de Azevedo-Santos, Maria do Desterro Soares Brandão Nascimento

**Affiliations:** 1Postgraduate Program in Northeast Biotechnology Network (RENORBIO), Federal University of Maranhão, São Luís 65080-805, Brazil; 2Postgraduate Program in Animal Science, State University of Maranhão, Sao Luis 65055-310, Brazil; 3Laboratory of Protozoology, Oswaldo Cruz Institute, Oswaldo Cruz Foundation, Rio de Janeiro 21041-250, Brazil; 4Laboratory for Applied Cancer Immunology, Biological and Health Sciences Center, Federal University of Maranhão, São Luís 65080-805, Brazil; 5Immunophysiology Laboratory, Biological and Health Sciences Center, Federal University of Maranhão, São Luís 65080-805, Brazil; 6Natural Products Department, Institute of Pharmaceutical Technology, Oswaldo Cruz Foundation, Rio de Janeiro 21040-900, Brazil; 7Electron Microscopy Nucleus, Adolfo Lutz Institute, Sao Paulo 01246-000, Brazil; 8Postgraduate Program in Adult Health (PPGSAD), Federal University of Maranhão, São Luís 65080-805, Brazil; 9Health Research Network, Research Center of Portuguese Oncology, Institute of Porto (CI-IPOP/RISE@CI-IPOP), Rua Dr. António Bernardino de Almeida, 4200-072 Porto, Portugal

**Keywords:** cancer, immunoprotection, *Euterpe oleracea*, açaí, antitumor activity, mouse model

## Abstract

**Simple Summary:**

Cancer is one of the major current public health threats worldwide that needs new forms of treatment. The seeds of *Euterpe oleracea* fruit, popularly known as açaí, are part of the fruit that are rejected during its processing for consumption, although the seeds have several compounds with pharmacological potential. This study aimed to evaluate the effect of açai seed extract against cancer cells in vitro and in vivo in animals with an Ehrlich tumor, an inflammatory mammary type of cancer. The açai seed extract displayed a rich chemical composition with a high content of catechin/epicatechin. Prostate cancer cell lines were killed by açai seed extract treatment in vitro. Animals with an Ehrlich tumor treated with açai seed extract showed a decrease in tumor size and an enhancement of the immunological response against the tumor. All these results point to a promising antitumor effect of açai seed extract that should be further clarified.

**Abstract:**

*Euterpe oleracea* (açaí) fruit has approximately 15% pulp, which is partly edible and commercialized, and 85% seeds. Although açaí seeds are rich in catechins—polyphenolic compounds with antioxidant, anti-inflammatory, and antitumor effects—almost 935,000 tons/year of seeds are discarded as industrial waste. This work evaluated the antitumor properties of *E. oleracea* in vitro and in vivo in a solid Ehrlich tumor in mice. The seed extract presented 86.26 ± 0.189 mg of catechin/g of extract. The palm and pulp extracts did not exhibit in vitro antitumor activity, while the fruit and seed extracts showed cytotoxic effects on the LNCaP prostate cancer cell line, inducing mitochondrial and nuclear alterations. Oral treatments were performed daily at 100, 200, and 400 mg/kg of *E. oleracea* seed extract. The tumor development and histology were evaluated, along with immunological and toxicological parameters. Treatment at 400 mg/kg reduced the tumor size, nuclear pleomorphism, and mitosis figures, increasing tumor necrosis. Treated groups showed cellularity of lymphoid organs comparable to the untreated group, suggesting less infiltration in the lymph node and spleen and preservation of the bone marrow. The highest doses reduced IL-6 and induced IFN-γ, suggesting antitumor and immunomodulatory effects. Thus, açaí seeds can be an important source of compounds with antitumor and immunoprotective properties.

## 1. Introduction

Cancer is one of the most well-known chronic non-communicable diseases of mankind, being the second leading cause of death in the world. In 2018, 18 million new cases of cancer and 9.6 million deaths occurred worldwide, and an estimated 25 million new cases are expected to occur in 2032 [1]. One of the main challenges in the treatment of cancer is that not all tumors respond to chemotherapy treatments, limiting the therapeutic effectiveness of these agents; in addition, many chemotherapeutic drugs cause serious adverse effects [2]. For that reason, there is an intense search for compounds, of natural or synthetic origin, that may prove to be a more effective alternative in chemotherapy treatment.

*Euterpe oleracea* Mart. is an Aracaceae native to the Amazon region, whose fruits are rich in polyphenols, compounds to which anti-inflammatory, antioxidant, cardioprotective, and anticancer activities are attributed [3,4]. The fruit of *E. oleracea*, known as açaí, has approximately 15% pulp, an edible part with important nutraceutical potential, and 85% seeds, which are neglected as industrial waste [5]. The açaí pulp, besides being highly nutritious, has been widely studied and has a variety of biological activities, including anti-inflammatory, antioxidant, cardioprotective, and anticancer effects [4,6,7], mainly attributed to its main constituents, the anthocyanins cyanidin-3-glycoside and cyanidin-3-rutinoside [8,9].

On the other hand, although açaí seeds are rich in flavonoids—mainly (−)-epicatechin, (+)-catechin, and (−)-epigallocatechin gallate [10], polyphenolic compounds capable of inducing antioxidant, anti-inflammatory, and antitumor effects [11]—thousands of tons (about 935,000 t/year) of seeds are normally discarded as “organic waste”, without an adequate economic destination. To our knowledge, there is only one study [12] showing the use of açaí seed flour in the prevention of obesity-induced hepatic steatosis, and few reports concerning the cytotoxic and pro-apoptotic effects of açaí seed extracts against tumor cell lines in vitro, such as breast (MCF7), prostate (DU145), lung (A549 and NCI-H460), liver (HepG2), and cervical (HeLa) cancer cells [11,13,14,15].

As far as we know, there are no reports on the antitumor properties of açaí seed extracts in vivo. This work aimed to study the antitumor effects of a hydroalcoholic extract of açaí seeds in vitro against prostate cancer LNCaP clone FGC cells, and in vivo using the Ehrlich tumor model in mice, which is associated with strong chronic inflammation. This model is widely used in the investigation of the mechanisms of tumor proliferation, as well as the host’s inflammatory and oxidative responses against tumor cells.

## 2. Materials and Methods

### 2.1. Plant Material

The specimens of *E. oleracea* were collected during 2018 March at Juçara Park, São Luís, Maranhão, Brazil (2°37′38.9″ S 44°17′28.9″ W). The plant material was authenticated by the Ático Seabra Herbarium of the Federal University of Maranhão, where a voucher specimen was registered (No. 01425). The access for the genetic heritage material was granted by the National System for the Management of Genetic Heritage and Associated Traditional Knowledge (Sistema Nacional de Gestão do Patrimônio Genético e do Conhecimento Tradicional Associado—SisGen) from the Ministry of the Environment of Brazil, voucher number A91B0BA.

### 2.2. Preparation of Hydroalcoholic Extracts from E. oleracea

Extracts from the whole fruit (called fruit here), the pulp, the seeds, and the hearts of palm (called palm here) were obtained. The *E. oleracea* fruits were cleaned and the seeds separated from the fruit in a Metvisa pulping machine, model DG10BIVMF60N51 (Metvisa, Brusque, Brazil). Then, the seeds were dried at room temperature (27–33 °C) for 3 days, after which the fibrous material was removed, and the seeds were ground in a grinder. The powder (360 g) was added in an ethanol/water solution (7:3; *v*/*v*), under continuous stirring by the Gehaka orbital AO–370 (Gehaka, São Paulo, Brazil) for 2 h, and allowed to macerate. The palm and the fruits were dried in an air circulation oven at 27 °C to 33 °C, 70% alcohol was added and left in the maceration for 15 days, and then it was filtered through Whatman No. 1 filter paper. The pulp was combined with PA ethanol, stirred for 2 h, then filtered through Whatman No. 1 filter paper. With the solid phase retained on the filter, the maceration was repeated for 48 h for 10 days, until the extracted solution became colorless. The filtered liquid phase of the palm, fruit, and seed were placed in an amber glass bottle. After macerating the solid phase, the mixed liquid phases were concentrated in a manufacturing microprocessed rotary evaporator at a bath temperature of 30 °C. In the end, the four concentrated extracts were dehydrated in a lyophilizer Liotop K105 (Liobras, São Carlos, Brazil) for 96 h at −101 °C and 23 mmHg, and kept at −20 °C until the day of use.

### 2.3. Characterization of Phenolic Compounds by ESI/MS and LC/MS-MS

The extracts were analyzed by direct infusion (ESI-MS/MS) in a Bruker Ion trap amazon SL mass spectrometer (Bruker, Billerica, MA, USA), positive (ESI+) and negative (ESI-) modes. Samples (2 mg) were dissolved using an ultrasonic bath in methanol, certified HPLC grade, for 20 min. The operating conditions were 1 µL/min infusion, 4.0 kV capillary voltage, 100 °C temperature source, and cone voltage of 20–40 V. Mass spectra were recorded and interpreted by Bruker Compass Data Analysis 4.2. The LC/MS-MS system was composed of an LC Shimadzu Nexera UFLC (Shimadzu Corp., Quioto, Japan) coupled to a Bruker Daltonics Amazon SL ion trap. The analysis was performed at room temperature in a 150 mm × 4.6 mm × 3.0 μm Waters Spherisorb S3 ODS-2 C18 column, using (A) 2.5% acetic acid in water and (B) HPLC-grade acetonitrile as the mobile phase. The analysis conditions and gradient followed those described by Barros et al. (2015) [11] for determining phenolic compounds of *E. oleracea*: 0% B for 5 min; from 0 to 10% B for 35 min; from 10 to 14.5% B for 5 min; from 14.5 to 19% B for 10 min; from 19% to 55% B for 10 min; isocratic 80% B for 3 min; and re-equilibration of the column, using a flow rate of 0.5 mL/min. The ion spray voltage was set at −4500 V in the negative mode.

### 2.4. Quantification of Catechin in the Hydroalcoholic Extract of E. oleracea Seeds

Catechin quantification in hydroalcoholic extract of *E. oleracea* seeds analyses were performed by HPLC/DAD-UV using a Shimadzu Nexera XR^®^ device, a liquid phase chromatograph coupled to a Shimadzu UV detector with a SPDM20A diode arrangement, equipped with a CBM20A controller, DGU20A degasser, LC20AD binary pump, CTO20A oven, and SILA20A auto-injector. The standard catechin solutions were prepared using 5.0 mg of catechin (SIGMA, Saint Louis, MO, USA) dissolved in 5.0 mL of methanol in a volumetric balloon, obtaining a final solution of 1 mg/mL. The preparation of the solutions for the analytical curve was performed by dilutions of the initial solution (1 mg/mL) in 10 mL of methanol (100 μg/mL). Successive dilutions of this solution yielded solutions with concentrations of 6.25, 12.5, 25, 50, 75, and 100 μg/mL. The solutions were filtered in a 0.45 μm PTFE filter (Merck Millipore, Ireland). 20 μL were injected in triplicate on 3 different days to obtain the analytical curves of the areas corresponding to the catechin peaks. The analytical curves (6.25–100 μg/mL) of catechin were constructed on the basis of the UV–Vis signal at 280 nm for better selectivity of their content, and the average of the 3 curves resulted in the following equation used in the calculation of the mg of catechin per g of dry extract: (g/mL) = (Abs (mAu)–6682)/4476; R^2^ = 0.9973). A Novapak C18 column was used (3.9 mm × 150 mm × 4 um); Flow: 1.3 mL/min; Oven temperature: 50 °C; Moving phases: A–acidified water pH 3.0; B–acidified water pH 3.0 + acetonitrile HPLC grade (50/50 *v*/*v*). Concentration gradient: 0.01 min: 12, 5% of solvent B; 5 min: 20% of solvent B; 8 min: 30% of solvent B, 17 min: 47% solvent B; 18 min: 100% solvent B; 25 min: 100% solvent; 27 min: 12, 5% solvent B; Time 30 min: 12, 5% solvent. Total running time: 30 min.

### 2.5. Cell Culture

Prostate adenocarcinoma cells DU 145 (ATCC HTB-81; BCRJ 0078) and LNCaP clone FGC (ATCC CRL-1740; BCRJ 0149), and uterine cervix carcinoma cell line HeLa (ATCC CCL-2; BCRJ 0100) were obtained from the Bank of Cells of Rio de Janeiro (BCRJ; http://bcrj.org.br/). The fibroblast CCD-1072Sk (ATCC CRL-2088; BCRJ 0062) was purchased from ATCC and is deposited in BCRJ by Dr. Katia Calabrese. The DU-145 and HeLa cells lines were cultured using Dulbecco’s Modified Eagle Medium (DMEM; Sigma-Aldrich, St Louis, MO, USA) supplemented with 10% fetal bovine serum (FBS; Gibco, Gaithersburg, MD, USA), 50 U/mL penicillin and 50 μg/mL streptomycin (Sigma-Aldrich, St Louis, MO, USA), and 2 mM L-glutamine (Invitrogen^®^, Waltham, MA, USA) in the culture medium. The cell line LNCaP clone FGC was cultured in RPMI 1640 (Sigma-Aldrich, St Louis, MO, USA), supplemented with 2 mM L-glutamine, 10% FBS, 100 U/mL penicillin and 100 μg/mL streptomycin, and 1.5 g/L sodium bicarbonate (Sigma-Aldrich, St Louis, MO, USA). Normal lung fibroblasts cell line (GM07492A) was cultured in enriched culture medium DMEM + Ham’s F10 Nutrient Mixture–F10 (1:1), supplemented with 10% FBS, 50 U/mL penicillin and 50 μg/mL streptomycin, and 2 mM L-glutamine, as per the recommendations of the Coriell Institute for Medical Research (https://www.coriell.org). The fibroblast CCD-1072Sk cells were cultured in Iscove’s modified Dulbecco’s medium (Sigma-Aldrich, St Louis, MO, USA), supplemented with 4 mM L-glutamine, 1.5 g/L sodium bicarbonate, 10% FBS, 100 U/mL penicillin, and 100 μg/mL streptomycin. All the cells were maintained in a CO_2_ (5%) stove at 37 °C during the growth and the experiments.

### 2.6. In Vitro Assay

The extracts were diluted in dimethyl sulfoxide (DMSO) (Thermo Fisher Scientific, Waltham, MA, USA) and then diluted in cell culture medium, with a DMSO final concentration < 0.01%. The solutions were filtered through a syringe filter with 0.2 μm pores and stored at −20 °C until use. Initially, the extracts were evaluated against the DU 145, HeLa, and GM07492A cell lines. Cells were treated with *E. oleracea* extract concentrations of 10–40 μg/mL for the viability assay with 3-(4,5-dimethylthiaxolone-2-yl)-2,5-diphenyl tetrazoliumbromide (MTT) (Sigma-Aldrich, Darmstadt, Germany). An aliquot of 1 × 10^6^ cells/mL was grown in 96-well microplates (100 µL/well), in the presence or absence of the extracts for 24 h. The supernatant was removed, the wells washed twice with phosphate-buffered saline, and 200 μL of DMEM containing 20 µL of MTT solution (5 mg/mL) was added in each well. The cells were incubated for 2 h in a CO_2_ incubator protected from light. After centrifugation of the plates at 240 g for 5 min at 4 °C, the supernatant was discarded, and the crystals were eluted in 100 µL ethanol P.A. The absorbance of the 540 nm wavelength through a microplate was determined with the microplate spectrophotometer reader EPOCH (Biotek Instruments, USA/Gen5 Software). After, *E. oleracea* extracts were evaluated against the LNCap clone FGC and the CCD-1072Sk cell lines. In 96-well plates, 100 µL per well of cell culture at 5 × 10^5^ cells/mL were incubated overnight at 37 °C and 5% CO_2_. Then, the medium was completely removed, and the cells were treated with 100 µL of *E. oleracea* extracts or docetaxel at different concentrations, obtained by serial dilution 1:2. The LNCaP clone FGC cells were treated with *E. oleracea* extracts concentrations ranging from 125 to 3.9 μg/mL. The CCD-1072Sk cells were treated with concentrations ranging from 1000 to 7.8 μg/mL. The reference drug docetaxel was evaluated with concentrations ranging from 100 to 0.78 μg/mL for both cells. Cells treated with DMSO 1% and wells with medium were kept as controls and blanks, respectively. After 24, 48, or 72 h, images of treated cells were acquired in light microscopy, and cell viability was determined with the sulforhodamine B (Sigma-Aldrich, St Louis, MO, USA) colorimetric method assay [16], with modifications. In brief, the cells were fixed with 10% trichloroacetic acid for 1 h at 4 °C, stained with 0.4% sulforhodamine B solution in 1% acetic acid for 30 min, and washed 3 times with 1% acetic acid solution. Sulforhodamine B was solubilized in 200 μL of 10 mM tris base solution, and the absorbance was determined in a spectrophotometer EZ Read 400 (Biochrom, Cambridge, UK) at a wavelength of 570 nm. The data were normalized, and cytotoxicity was demonstrated as a percentage corresponding to the viability calculated as described elsewhere [17], and the cell cytotoxicity by 50% (CC_50_) was obtained with GraphPad Prism 7.00.

### 2.7. Transmission Electron Microscopy

The LNCaP clone FGC cells were treated for 48 h with CC_50_ of *E. oleracea* fruit and seed extracts. Untreated cells were used as a control. After treatment, the cell layer was dissociated with trypsin-EDTA solution, and the cell suspension was fixed with 2.5% glutaraldehyde in a 0.1 M sodium cacodylate buffer, pH 7.2, overnight. Then, the cells were postfixed with 1% osmium tetroxide solution with 0.8% ferrocyanide and 5 mM calcium chloride, dehydrated in graded acetone, and embedded in EMBed-812^®^ resin (Electron Microscopy Sciences, Hatfield, PA, USA). Ultrathin 100 nm cuts were obtained in a Sorvall MT 2-B (Porter Blum) ultramicrotome (Sorvall, Newtown, CT, USA), stained with a 5% uranyl acetate aqueous solution and lead citrate (1.33% lead nitrate and 1.76% sodium citrate), and observed with a JEM-1011 transmission electron microscope (JEOL, Tokyo, Japan), operating at 80 kV [18].

### 2.8. Animals

A total of 36 male Swiss mice, with 8 weeks of age, weighing on average 35 g, were obtained from the central animal facility of the Federal University of Maranhão (UFMA) in São Luís, MA, Brazil, and maintained at 26 ± 2 °C, 44–56% relative humidity, under 12 h light–dark cycles, and with free access to sterile food and water. The study was conducted after approval by the Research Ethics Committee for the Use of Animals from UFMA (CEUA: N° 01.0341.2014.).

### 2.9. Ehrlich Solid Tumor Model

Ehrlich’s ascitic tumor, derived from a spontaneous murine mammary adenocarcinoma, was maintained in the ascitic form by passages in Swiss mice, by weekly transplantation of 10^6^ tumor cells i.p. [19]. The ascitic fluid was removed by opening the abdominal cavity and carefully collecting all the fluid with a sterile 10 mL syringe. Ascitic tumor cell counts were done in a Neubauer hemocytometer, presenting more than 99% of viable cells by a Trypan blue exclusion method. The final concentration of tumor cell suspension was adjusted to 2 × 10^6^ viable cells/mL. Finally, a volume of 0.05 mL of Ehrlich tumor cell suspension was injected into the foot pad to implant the solid form of the tumor.

### 2.10. Treatment and Tumor Development Assay

To treat the mice, the *Euterpe oleracea* seed extract was diluted in isotonic phosphate buffered solution (PBS) and administered by the oral route at doses of 100 mg/kg, 200 mg/kg, and 400 mg/kg (SE100, SE200, and SE400, respectively) body weight, daily, 24 h after Ehrlich tumor implantation. The negative control group (CTL−) was treated only with PBS, and the positive control group (CTL+) was treated with cyclophosphamide at a dose of 20 mg/kg body weight. The treatments were administered by gavage 15 days after the tumor cells inoculation. The parameters of tumor-free animals were used as a normal reference (Sham). The mice were weighed at day 0 and day 15 post-inoculation to assess weight gain. At day 16, the mice were euthanized with an overdose of anesthetic using 150 mg/kg ketamine hydrochloride and 120 mg/kg xylazine hydrochloride. Serum was collected, and tumor growth and immunological parameters were evaluated as described previously [20]. To evaluate tumor development, the tumor volume was determined by the difference between the measured volume and the basal volume, and evaluated every 4 days, up to 15 days after inoculation. After euthanasia, both legs were removed and weighed, providing the relative weight of the tumor, calculated as the difference between the weights of the tumor and healthy paws.

### 2.11. Immunological Organs Evaluation and Cytokines Detection

To obtain the immunological parameters, the organs were harvested aseptically and gently crushed in RPMI medium. To obtain bone marrow cells, the femur was infused with 1 mL PBS. To determine the cellularity of the lymph node, spleen, and bone marrow, 90 μL of the cell suspension was removed, and 10 μL of crystal violet was added. The cells were counted in a Neubauer chamber with a common light optical microscope [21]. The quantification of the cytokines Interleukin-6 (IL-6), Interleukin-10 (IL-10), Monocyte Chemoattractant Protein-1 (MCP-1), Interferon-γ (IFN-γ), Tumor Necrosis Factor alpha (TNF-α), and Interleukin-12p70 (IL-12p70) protein in serum was performed using the Cytometric Beads Array (CBA), according to the manufacturer’s description (Becton Dickinson, San Diego, CA, USA)

### 2.12. Histological Analyses

The tumor, footpad, spleen, liver, and kidneys were removed from all animals, weighed on a precision electronic scale, fixed in 10% formaldehyde (*v*/*v*), and processed by the routine paraffin inclusion technique. Sections of each tissue were stained using the hematoxylin eosin (HE) technique for histological analysis in bright-field optical microscopy Nikon Eclipse E200 (Nikon Instruments Inc., Tokyo, Japan). The slides were photographed using specific software (Aperio, Leica). In kidney tissues, the presence of tubular degeneration was diagnosed based on the presence of pallor and cytoplasmic tumefaction, karyomegaly, and necrosis of isolated tubular cells. In liver tissues, the presence of vacuolar hepatocellular degeneration was characterized as mild (restricted to periportal and/or centrilobular areas) or moderate (extending to midzonal areas); hepatitis was characterized as mild (Küpfer cells hyperplasia and/or occasional microabscesses or slight, focal periportal leukocytic infiltration) or moderate (multifocal to diffuse, leukocytic infiltration in multiple portal spaces or centrilobular veins); hepatocellular necrosis was characterized as present or absent; hepatocellular karyomegaly was characterized as more than 10% of hepatocytes with nuclei twice the size of normal hepatocytes; and the number of mitosis figures were characterized per 400x field. In the tumor samples, it was characterized by the presence of nuclear pleomorphism in tumor cells (defined as absent 0− regular nuclei; mild +− minimal differences between nuclei; moderate ++− changes involving the nuclear size and shape or nucleoli; or intense +++− marked changes in the nuclear size, shape, or coloration or nucleoli with the presence of bizarre nuclei), inflammatory infiltrate (cell types present, distribution: peritumoral, intratumoral, multifocal, diffuse, and intensity: absent 0, minimal +, moderate ++, intense +++), the pattern of the tumor growth and invasiveness (presence or absence of well-defined border) and their presence.

### 2.13. Statistical Analysis

Data were reported as the mean ± standard deviation (SD) regarding the normal data distribution analyzed by the Shapiro–Wilk normality test. Statistical analyses were performed using Graph Pad Prism version 7.0 software with ANOVA, with Tukey’s post-hoc test and the Kruskal–Wallis test multiple comparison for cytokines analysis. We considered a *p* value lower than 0.05 as statistically significant. For each set of data, the results correspond to the average ± standard deviation of at least two independent experiments, with each one performed at least in triplicate.

## 3. Results

### 3.1. Characterization of Phenolic Compounds and Catechin Quantification in E. oleracea Extracts

The general characterization of the extracts was obtained by ESI/MS analyses. The comparison between the tentatively identified compounds of each extract is shown in Table 1.

Compounds 5 (tR = 15.5 min) and 6 (tR = 2.7 min) exhibited the molecular ion ([M]-) at *m*/*z* 288.99 and *m*/*z* 289.03 and were identified as (+)-catechin and (−)-epicatechin, respectively, by comparing their retention time and molecular ions with commercial standards. Other compounds were tentatively identified by their similarity to literature data [5,11] and are presented in Table 2. In the table, according to the relative peak area, (+)-catechin was the major compound (45.81%), followed by caffeic acid-3-glucoside (23.91%) and (−) epicatechin (6.02%). The chromatograms obtained from the HPLC–MS/MS of the hydroalcoholic extract of the seeds and fruit of *E. oleracea* are presented in Figure 1. The quantification of the catechin in the hydroalcoholic extract from the seeds of *E. oleracea* was performed as described in item 2.4 of the experimental section, and the results in mg of catechin per g of extract was 86.26 ± 0.189.

### 3.2. In Vitro Antitumor Effect of E. oleracea Extracts

Extracts were evaluated first after 24 h of treatment, and the cytotoxicity was analyzed in 2 tumor cell lines, human prostate cancer DU-145 (Figure 2A) and human cervical cancer HeLa (Figure 2B), and in the non-tumoral cell line GM5849, immortal human fibroblast (Figure 2C). The extracts did not show cytotoxic effects after 24 h of treatment, but the cell viability was lower in the tumor line cell treated with seed extract when compared with the normal cells, suggesting a cytostatic effect for these cells. The extracts were then evaluated against the LNCaP clone FGG cell line, a human prostate cancer that is hormone receptor-positive, and the CCD-1072Sk cell line, a human fibroblast. The only extracts that showed activity against the LNCaP cells were the fruit and seed extracts (Table 3). The pulp and palm extracts did not demonstrate cytotoxicity to LNCaP or CCD-1072Sk in the concentrations analyzed (IC_50_ > 1000 µg/mL). The reference drug docetaxel showed antitumor and cytotoxic activity, as expected.

The seed extract exhibited the best antitumor results against LNCaP cells, and its inhibitory effect was not time-dependent, considering the times of the treatment analyzed (Figure 2E). Likewise, the seed extract also displayed the highest toxicity against CCD-1072Sk cells, and its effect was time-dependent (Figure 2G). The fruit extract exhibited the same pattern (Figure 2D,F), although with higher values than the seed extract for all the parameters evaluated. The maintenance of antitumor activity against LNCaP cells associated with the increase in toxicity against CCD-1072Sk by the seed extract resulted in a decrease in the selectivity index (SI) over time; however, the SI values remained higher than tenfold. The effect of *E. oleracea* extracts treatment in LNCaP cells is remarkably visible by light microscopy, with both the seed and fruit extracts inducing vesicles of pronounced size that led to the loss of the characteristic structure of the cell (Figure 2H).

In the same way, ultrastructural alterations were similar for both extracts, with the fruit extract treatment (Figure 3B–F) inducing less severe ultrastructural changes than the seed extract (Figure 3G–K). The main alterations observed by treatment was mitochondrial alteration, such as a decrease in size and shape, swelling, loss of mitochondrial cristae, and dissolution of the mitochondrial matrix; the presence of numerous residual bodies; and the alteration of chromatin and pyknotic nuclei. Untreated cells showed normal morphology (Figure 3A).

### 3.3. E. oleracea Seed Extract Reduced Ehrlich Solid Tumor Growth

The tumor-inoculated paw’s weight was higher in tumor-bearing mice compared with tumor-free animals, demonstrating the effectiveness of the tumor model. Thus, the weight of the paws of the animals was measured in order to determine whether the treatment with the extract would be effective in reducing tumor growth. As expected, chemotherapeutic treatment reduced the paw’s weight, showing no difference from the sham group. On the other hand, *E. oleracea* seed extract at 100 (SE100) and 200 (SE200) mg/kg did not induce significant differences when compared with the CTLO group. However, the dose of 400 mg/kg (SE400) significantly reduced the paw’s weight, similar to the group treated with cyclophosphamide (CTL+) (Figure 4A). In accordance with these data, treatments with 200 and 400 mg/kg of *E. oleracea* seed extract and cyclophosphamide inhibited tumor growth, as compared to the negative control (CTL−), after the fourth day and maintained it through the end of the experiment (Figure 4B); these data were corroborated by the area under the curve values (Figure 4C). This result clearly demonstrates that, at a dose of 400 mg/kg, açaí seed extract was as effective as the dose of cyclophosphamide (20 mg/kg) in reducing tumor growth. Histological analyses of the tumor tissue evaluated nuclear pleomorphism, necrosis, mitotic figures, and infiltrated inflammatory cells. The analyses showed marked pleomorphism and inflammatory infiltrates in the Ehrlich solid tumor (Figure 5).

The cyclophosphamide (CTL+) induced lower histological scores compared with the CTL−. *E. oleracea seed* extract treatment showed an intermediate behavior, but the 400 mg/kg (SE400) group achieved the lowest score for mitotic figures and the highest for inflammation (Table 4).

### 3.4. Immune Preservation Effects of E. oleracea Extract

Ehrlich’s solid tumor induced an increase in the weight and the cellularity of the spleen and the popliteal lymph node, suggesting inflammation or metastasis towards lymphoid organs. However, the *E. oleracea* seed extract reduced the spleen and popliteal lymph node weight and cell numbers. The results for the lymph node weight at 400 mg/kg were similar to those obtained with cyclophosphamide, but the same was not observed for the bone marrow (Table 5). Cyclophosphamide induced degenerative changes in the kidney and liver, but *E. oleracea* seed extract was not associated with significant renal or hepatic histological changes (Table 6). The quantification of cytokine levels is presented in Figure 6. The tumor upregulated pro-inflammatory cytokines TNF-α (Figure 6A), MCP-1 (Figure 6B), and IL-6 (Figure 6C) when compared with the sham group. The treatment with cyclophosphamide and *E. oleracea* seed extract (400 mg/kg) downregulated most inflammatory cytokines, but upregulated IFN-γ production (Figure 6D). IL-12p70 and IL-10 were not detected. The data suggest that the seed extract treatment modified the cytokine’s pattern, supporting the cytotoxic immune response increased by IFN-γ and reducing the chronic inflammation increased by IL-6 and MCP-1 modulation.

### 3.5. E. oleracea Seed Extract Treatment Did Not Show Toxicity

To evaluate if oral treatment with seed extract can influence the weight of animals throughout the experiment, at the end of the fifteenth day of experiments, the weight gains (g) of the animals in the Sham (40.05 ± 0.77), CTL− (34.97 ± 0.44), CTL + (38.88 ± 0.57), SE100 (32.65 ± 0.69), SE200 (29.48 ± 0.9), and SE400 (28.58 ± 1.17) groups were not statistically different. Furthermore, the results of histological analysis showed that the seed extract treatment did not change the structure of the liver and kidney tissue. The data indicate the absence of toxic symptoms from the seed extract in the treated animals.

## 4. Discussion

Cancer treatment is one of the most challenging areas of current medicine, and the need for new options to integrate the limited arsenal of antitumoral agents is urgent. This work aimed to assess the antitumoral effect of *E. oleracea* extracts against different cell lines in vitro, as well as to evaluate the effect of *E. oleracea* seed extract in the solid Ehrlich carcinoma model.

The chemical profile of the four extracts obtained in the study was derived by LC/MS analyses and allowed the identification of the substances by comparison with literature data, based both on the molecular masses found and on the retention times observed in each chromatogram. The evaluation of the anthocyanidin composition in the different açaí extracts demonstrated variability, both in the type and in the concentration of these substances according to the part of the plant, corroborating previous reports of the *E. oleracea* chemical profile [13,22]. In the chromatographic analysis of açaí seed extract, the major compounds identified were (+)-catechin, caffeic acid-3-glucoside, and (−)-epicatechin, compounds whose antioxidant, anti-inflammatory, and anticarcinogenic activities have been reported [11,23,24]. The chemical compound can be directly associated with its inhibitory effect in cancer cell lines in vitro.

For all the extracts evaluated in vitro, the seed extract exhibited promising results. The açaí seed extract prevented the proliferation of cancer cells from cervical cancer (HeLa) and prostate cancer (DU145), regardless of the concentration. The inhibitory activity assays against the prostate cancer cell line LNCaP clone FGC displayed remarkable activity of the fruit and seed extracts. The antitumoral effects of açaí seed extract have also been reported in breast, colon, and lung human cancer cell lines, where it induced apoptosis, autophagy, and cell cycle arrest [11,13,14,25]. Further, the seed and fruit extracts were the only two that showed the presence of (+)-catechin and (−)-epicatechin, which should be related to the antitumoral activity observed in our study and in other studies reported in the literature [26,27,28]. Docetaxel is a well-known therapeutic option in the treatment of prostate cancer [29], with a direct effect on tubulin (no need for metabolic conversion), allowing its use in in vitro assays [30]. In our experiments, docetaxel exhibited activity against the prostate cancer cell line LNCaP clone FGC, as expected.

The extracts were not cytotoxic to healthy fibroblast cells (GM5984), but showed a concentration-dependent and time-dependent toxicity against fibroblast CCD-1072Sk. However, the selectivity index of the seed extract remained higher, even after 72 h of treatment. This result is very encouraging, demonstrating the seed extract’s safety and its pronounced selectivity against the prostate cancer LNCaP cell line. It has been reported that phenolic compounds exert antioxidant actions on normal cells and pro-oxidant actions on tumor cells. The pro-oxidant actions of phenolic compounds are based on the generation of a phenoxyl radical or a redox complex with a transition metal ion, which generate reactive oxygen species (ROS), causing oxidative damage [31]. As cancer cells have a higher concentration of copper ions and greater metabolic activity than normal cells, pro-oxidant compounds increase cellular levels of ROS to cytotoxic levels in cancer cells, but not in normal cells [32].

Treatment of LNCaP cells with seed and fruit extracts caused alterations observable by light microscopy and transmission electron microscopy. The large vacuoles induced by the treatment were observed by both methodologies and altered the cell morphology, as shown by light microscopy. Besides the residual bodies observed by transmission electron microscopy, several mitochondrial alterations were observed after treatment with the fruit and seed extracts. The ultrastructural mitochondria changes ranged from the modifications in size and shape due to swelling and loss of mitochondrial cristae to the complete degeneration of the organelle, characterizing mitochondrial disfunction produced by *E. oleracea* treatment. A noteworthy alteration in chromatin condensation and pyknotic nuclei was also observed after treatment. The association between mitochondrial dysfunction, generation of ROS, and chromatin changes with autophagy is well reported [33,34], and the ultrastructural changes observed after treatment with *E. oleracea* indicate a mechanism of action in this sense. Nevertheless, there are reports in line with our results that *E. oleracea* seed extract induced ROS production and increased mitochondria numbers, suggesting ROS enrollment in autophagy and cell death, possibly by autophagy, which were confirmed by the autophagolysosome formation in the MCF-7 breast cancer cell line [35,36]. Another study demonstrated a reduction in MCF-7 cell viability and death by necroptosis after *E. oleracea* treatment, revealing that the in vitro mechanism of action still needs enlightenment.

In the same manner, although there are reports of photodynamic therapy mediated by acai oil in a nanoemulsion against melanoma in a murine model [37], and even a phase II clinical trial of a commercial acai juice product in biochemically recurrent prostate cancer [38], the in vivo anticancer activity of the *E. oleracea* seed extract remained unknown. The present study makes an important contribution by reporting the efficacy of this extract against an in vivo breast cancer model based on Ehrlich’s solid tumor. The results show that *E. oleracea* seed extract was able to partially block tumor growth when administered at the highest dose. Interestingly, the present extract reduced the pleomorphism and mitotic activity of cancer cells, while increasing the presence of necrosis and immune inflammatory cells in tumor tissue. These data suggest that, in addition to the mechanisms described in the induction of direct cell cycle arrest and cell death, the extract may have an immunogenic effect by improving the host response against the tumor. Tumor-associated inflammation and immunity is highly complex and can lead to cancer control and even eradication, but they can also be hijacked by cancer cells to promote cancer progression [39]. The inflammatory microenvironment generated in tumor stroma includes multiple types of leukocytes (e.g., tumor-associated macrophages, TAMs), which express cytokines, chemokines, growth factors, and proteases and can enhance tumor progression or mediate antitumor responses [40]. Modulating inflammation and the immune response is a valid strategy to prevent and treat various types of cancer, and it is tempting to speculate that the inflammatory modulation induced by *E. oleracea* seed extract may, at least partly, explain its antitumoral effect. Cyclophosphamide is a drug that reduces the risk of recurrence and mortality in early breast cancer [41]. Although the use of cyclophosphamide in vitro would be unfeasible, since its action depends on a primary conversion that occurs in the liver [42], it modulates inflammation and the immune response in the Ehrlich model, and is used as a reference drug in in vivo breast cancer experimental models [20,43,44,45].

In this study, *E. oleracea* seed extract significantly changed the circulating cytokine profile of treated animals and demonstrated a marked capacity to reduce the production of inflammatory cytokine IL-6 in a dose-dependent manner, to modulate the production of TNF-α, and to partially modulate the production of the chemokines MCP-1 and IFN-γ. The production of IL-6 is strongly related to the progression and development of several different types of tumors, possibly due to its ability to promote the growth of tumor cells and to favor angiogenesis in an inflammation-dependent context [46]. In many cases, the plasma levels of this cytokine are directly related to the aggressiveness of the disease and a poor prognosis [47]. Counterintuitively, the same response pattern is observed in relation to TNF-α. Although this cytokine has antitumor activity, its production in large quantities and/or in a constant manner can lead to tumor proliferation, as well as angiogenesis and metastases, demonstrating in a representative way the tumor’s escape ability in relation to the immune system and the dangers of a non-resolving inflammatory process [48].

The production of the chemokine MCP-1 also demonstrates this unfavorable modulation of the immune system, since the interaction of TAMs can produce this mediator to recruit other monocytes for the microenvironment, and once there, produce inflammatory cytokines (among them IL-1, IL-6, and TNF-α) that end up promoting the tumor progression [49]. Thus, the ability of the highest doses of the extract to decrease these three cytokines, without attempting to deplete them, demonstrates an ability to restore the immune balance in favor of tumor elimination, in addition to confirming the anti-inflammatory properties of *E. oleracea*.

This hypothesis is supported when we observe that in the highest dose, the extract is able to favor the production of IFN-γ, demonstrating a clear tendency for the Th1 response, where the activation of the cellular immune response allows the destruction of tumor cells through mechanisms dependent on (CD8 +) and independent of (NK, NKT) the presentation of the MHC class I [50,51]. Beyond that, the presence of IFN-γ promotes a more favorable activation for the elimination of cancer cells by the macrophages (M1 phenotype) present in the tumor microenvironment [52]. This theoretical antitumor action is strongly corroborated by the reduction of the tumor size, the greater presence of necrosis, and the fewer mitosis numbers in the group treated with the three concentrations of the extract.

Another set of findings is related to preliminary toxicological analyses to determine the safety of the *E. oleracea* seed extract. In the literature, there are studies using 300 mg/kg/day of açaí seed extract rich in proanthocyanidins for 12 weeks [53], and our group already used a single treatment dose of 1000 mg/kg of catechin-rich açaí seed extract similar to that used in this research [22], with the observation of no toxicity. In addition, catechin, the major compound in açaí seed extract, also has studies reporting its low potential for toxicity induction. In PubChem, the LD50 of catechin administered by the oral route is >10 g/kg in mice. A preliminary in silico evaluation using pkCSM [54], performed by us, predicted no hepatotoxicity and an Oral Rat Chronic Toxicity (LOAEL) of 316.2 mg/kg bw/day of catechin. Considering the catechin concentration of approximately 45% in our extract, it allows the use of a dose of up to 702.6 mg/kg bw/day of açaí seed extract. In this context, the weight gain and histological analyses of hepatic and renal tissues did not evidence any toxic changes related to the administration of the *E. oleracea* seed extract, even at the highest dosage. This suggests that the extract presents a favorable toxicological profile, but further studies are needed to characterize its toxicity in multiple organs at the morphological and biochemical levels.

## 5. Conclusions

The seed extract of *E. oleracea* presented a higher catechin/epicatechin content than the fruit, palm, and pulp extracts. The seed extract also exhibited the best activity in vitro against the LNCaP prostate cancer cell line, inducing mitochondrial and nuclear alterations. The *E. oleracea* seed extract showed antitumor activity on in vitro and in vivo assays, which can be attributed to its rich composition in phenolic compounds. The present study proposes that the extract has immunogenic and anti-inflammatory properties that contribute to its antitumor activities. These data may be important for identifying potential compounds in a part of the fruit of the species that is considered waste, and possibly helping to provide subsidies for the valorization of this resource and a possible new source of income for economic purposes.

## 6. Patents

The patent BR10201901353 “Formulações farmacêuticas com extrato da semente de *Euterpe oleracea* Mart.” was granted to Muniz Filho, W. E.; Nascimento, M. D. S. B.; and Santos, A. P. S. A. by the National Institute of Industrial Property (Instituto Nacional da Propriedade Industrial—INPI) in 2019/06/28.

## Figures and Tables

**Figure 1 cancers-15-02544-f001:**
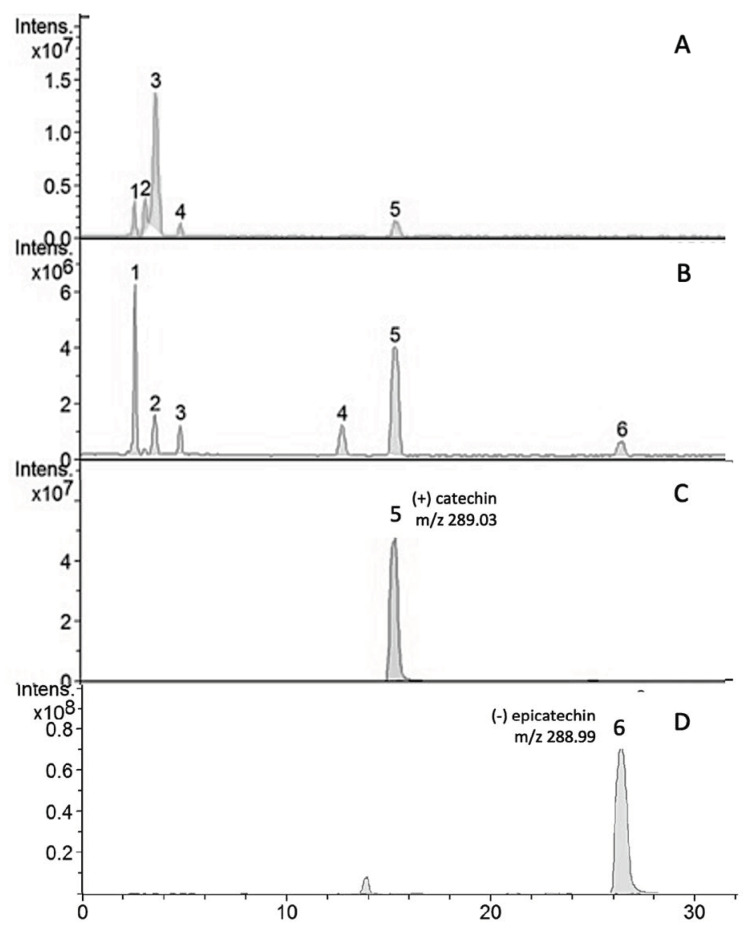
HPLC-MS/MS phenolic profiles of hydroalcoholic extracts of (**A**) *Euterpe oleracea* fruit, (**B**) *Euterpe oleracea* seeds, (**C**) (+) catechin standard, and (**D**) (−) epicatechin standard.

**Figure 2 cancers-15-02544-f002:**
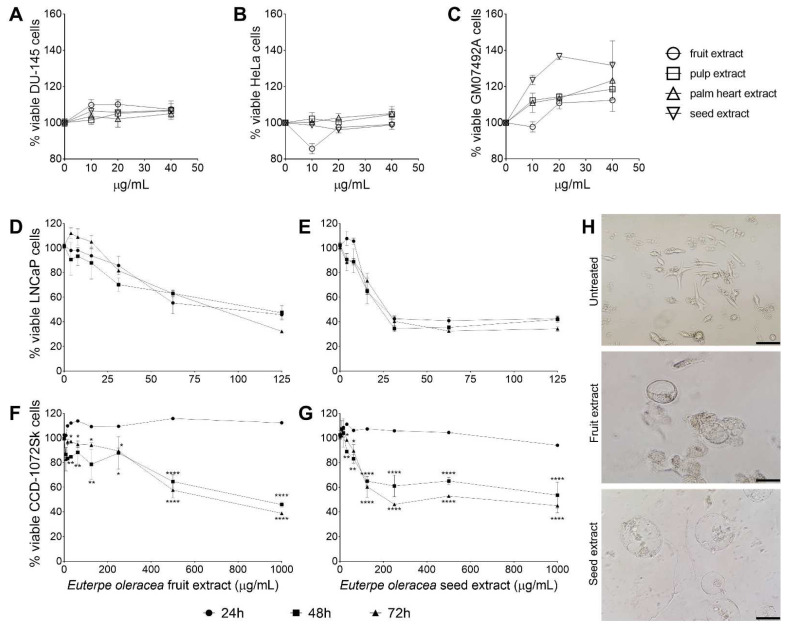
*Euterpe oleracea* antitumoral activity in vitro. Activity of *E. oleracea* against DU-145 (**A**), HeLa (**B**), and GM07492A (**C**) cells. Concentration response curve of LNCaP clone FGC (**D**,**E**) and CCD-1072Sk (**F**,**G**) cells treated with fruit and seed extract of *E. oleracea*. Data represent the mean ± standard error of mean of experiment carried out in triplicate. * *p* < 0.05, ** *p* < 0.01, **** *p* < 0.0001, when compared with 24 h-treated cells by two-way ANOVA and Tukey’s multiple comparisons tests. (**H**) Light microscopy of LNCaP clone FGC cells treated for 72 h with *E. oleracea* extracts. Bar scale: 150 µm.

**Figure 3 cancers-15-02544-f003:**
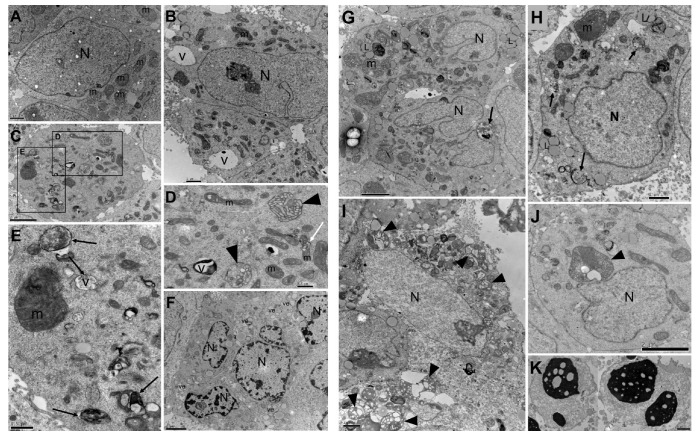
Transmission electron microscopy of LNCaP clone FGC cells treated for 48 h with CC_50_ of fruit (**B**–**F**) and seed (**G**–**K**) *Euterpe oleracea* extracts. (**A**) Untreated cells. (**B**–**E**) Presence of large vacuoles (V), residual bodies (arrows); alteration in size and shape of mitochondria; swelling and loss of mitochondrial matrix (head arrows). (**F**) Cells with numerous vesicles (ve) and alteration of chromatin condensation in nuclei. (**G**–**J**) Presence of lipids (L), residual bodies (arrows), increase in the number of mitochondria with different shapes and sizes; mitochondria swelling, and loss of mitochondria cristae and matrix (head arrows). (**K**) Pyknotic nuclei. N: nucleus, m: mitochondrion. Scale bar is 0.5 µm in (**D**,**E**); 1 µm in (**A**,**B**,**H**,**I**,**K**); and 2 µm in (**C**,**F**,**G**,**J**).

**Figure 4 cancers-15-02544-f004:**
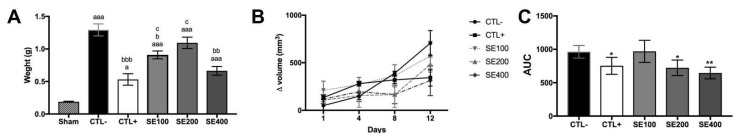
Antitumor effect of *Euterpe oleracea* seed extract in in vitro and in vivo assays. (**A**) Tumor weight reduction of the Ehrlich solid tumor due to treatments with açaí seed extract and cyclophosphamide. (**B**) Kinetic curve of paw volume during the tumor development. The groups treated with seed extract at 100 mg/kg (SE100), 200 mg/kg (SE200), and 400 mg/kg (SE400) were compared with the group without solid tumor (Sham), represented by “a”; the tumor-inoculated group treated with saline (CTL−) is represented by “b”, and the tumor-inoculated group treated with cyclophosphamide at 20 mg/kg (CTL+) is represented by “c”. The statistical difference was represented by one letter for *p* < 0.05, two letters for *p* < 0.01, and three letters for *p* < 0.001 (ANOVA and Tukey’s post-hoc test). (**C**) Area under the curve (AUC) for each individual tumor growth curve shown in Figure 3B. Data represent the mean ± S.D. * *p* < 0.01 and ** *p* < 0.001 compared with negative control group (CTL−) (ANOVA and Tukey´s post-hoc test).

**Figure 5 cancers-15-02544-f005:**
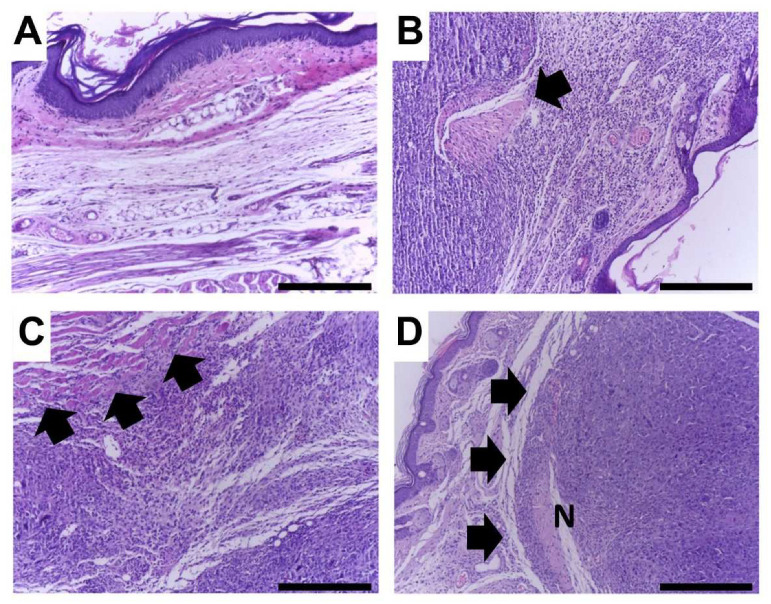
Representative histological images of treated and untreated tumor lesions, H&E, 40×. Scale bar = 300 µm. (**A**) Normal histology in a mouse without tumor induction. (**B**) Ehrlich tumor in an untreated mouse; note marked tumor invasion with destruction of skeletal muscle (arrow). (**C**) Ehrlich tumor in a cyclophosphamide-treated mouse; note marked tumor invasion with destruction of skeletal muscle (arrows). (**D**) Ehrlich tumor in a mouse treated with 400 mg/kg extract; note well-delimited tumor with a compressive growth pattern (arrows). A necrotic area is denoted by the letter N.

**Figure 6 cancers-15-02544-f006:**
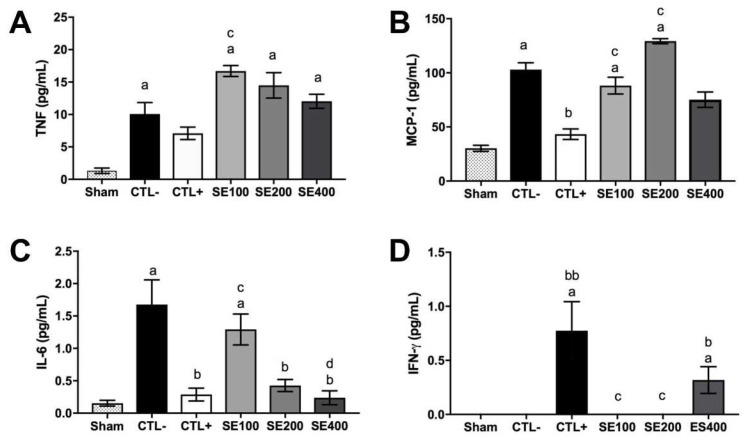
Cytokines modulation effect of *Euterpe oleracea* seed extract of Ehrlich solid tumor model. Levels of TNF-α (**A**), MCP-1 (**B**), IL-6 (**C**), and IFN-γ (**D**) in animal serum. Groups treated with seed extract at 100 mg/kg (SE100), 200 mg/kg (SE200), and 400 mg/kg (SE400) were compared with the group without solid tumor (Sham) represented by “a”; the tumor-inoculated group treated with saline (CTL−) represented by “b”, and the tumor-inoculated group treated with cyclophosphamide at 20 mg/kg (CTL+) represented by “c”. Data represent the mean ± SEM. The statistical difference was represented by one letter for *p* < 0.05 and two letters for *p* < 0.01 (Kruskal–Wallis multiple comparisons test).

**Table 1 cancers-15-02544-t001:** Mass spectrometry of *Euterpe oleracea* extracts from pulp, palm, seeds, and fruit.

*m*/*z* Exp	Mode	Molecular Formula	*E. oleracea* Extracts				Tentative Identification
			Pulp	Palm	Seeds	Fruit	
289.0669	neg	C_15_H_14_O_6_	−	−	+	+	Catechin/epicatechin
299.0507	neg	C_16_H_12_O_6_	+	+	−	+	Kaempferide
433.1364	pos	C_21_H_21_O_10_	+	−	+	+	Pelargonidin 3-O-glucoside
449.1192	pos	C_21_H_21_O_11_	+	−	+	−	Cyanidin-3-O-galactoside
461.1047	neg	C_22_H_21_O_11_	−	+	−	+	Chrysoeriol 7-O-glucoside
463.1503	pos	C_22_H_23_O_11_	+	−	−	−	Peonidin 3-O-glucoside
595.1843	pos	C_27_H_31_O_15_	+	−	−	−	Cyanidin 3-O-rutinoside
609.1869	pos	C_28_H_33_O_15_	+	−	−	+	Peonidin 3-O-rutinoside
865.1835	neg	C_45_H_38_O_18_	−	−	+	−	Procyanidin trimer C1

*m*/*z* exp: mass/charge experimental; *m*/*z* calcd: mass/charge calculated. Signals (+) and (−) indicate the presence or absence of the compound.

**Table 2 cancers-15-02544-t002:** Retention time (Rt), pseudomolecular and MS^2^ fragment ions, and tentative identification of phenolic compounds in *Euterpe oleracea* seed extract.

Compound	Rt (min)	Molecular Ion [M-H]^−^ (*m*/*z*)	MS^2^ (*m*/*z*)	Area (%)	Tentative Identification
**1**	2.7	341.16	280/178	23.91	Caffeic acid-3-glucoside
**2**	3.7	267.01	206/160	9.46	n.i.
**3**	5.0	267.00	206	5.03	n.i.
**4**	12.9	577.21	559/451/425/288/244	9.74	B-type (epi) catechin dimer
**5**	15.5	288.99	244/204	45.81	(+) catechin
**6**	26.7	289.03	244/204/124	6.02	(−) epicatechin

n.i. = not identified.

**Table 3 cancers-15-02544-t003:** Cytotoxic concentration for 50% and selectivity index of LNCaP cells clone FGC and CCD-1072Sk after 24, 48, and 72 h of treatment with *Euterpe oleracea* extracts and docetaxel.

Extract/ Compound	LNCaP clone FGC CC_50_ (µg/mL)			CCD-1072Sk CC_50_ (µg/mL)			SI		
	24 h	48 h	72 h	24 h	48 h	72 h	24 h	48 h	72 h
Fruit	>125	98.20 ± 1.219	119.5 ± 1.470	>1000	859.3 ± 1.245	844.7 ± 1.184	>7.68	8.75	7.06
Seed	44.0 ± 1.225	32.70 ± 1.226	30.80 ± 1.208	>1000	650.8 ± 1.251	402.9 ± 1.215	>22.72	19.90	13.08
Docetaxel	1.52 ± 0.021	1.28 ± 0.055	1.34 ± 0.041	70.16 ± 1.264	45.52 ± 1.276	36.48 ± 1.258	46.15	35.56	34.44

CC_50_: cytotoxic concentration of 50% cells; SI: selectivity index calculated from the ratio of CC_50_ values of CCD-1072Sk/LNCaP clone FGC.

**Table 4 cancers-15-02544-t004:** Histological scores in tumor tissue and matched normal tissues in all experimental groups.

Histological Features	Scores	Animal Groups					
		Sham	Ehrlich+				
			CTL−	CTL+	SE (mg/kg)		
					100	200	400
Nuclear pleomorphism	0	6	-	-	-	-	-
	+	-	-	-	-	-	-
	++	-	-	4	3	1	2
	+++	-	6	2	3	3	3
Necrosis	0	6	-	-	-	-	-
	+	-	2	-	-	-	-
	++	-	3	2	2	2	4
	+++	-	1	4	4	2	1
Mitotic figures	0	6	-	-	-	-	-
	+	-	-	-	-	-	-
	++	-	2	5	4	2	5
	+++	-	4	1	2	2	-
Inflammatory infiltrate	0	6	-	-	-	-	-
	+	-	-	-	-	-	-
	++	-	3	2	3	-	1
	+++	-	3	4	3	4	4
Tumor invasion	0	6	-	-	-	-	-
	+	-	3	-	1	-	-
	++	-	1	2	-	-	2
	+++	-	1	4	5	4	3

Mice without tumor induction (Sham; *n* = 6); mice with tumors treated with saline (CTL−; *n* = 6); mice with tumors treated with cyclophosphamide (CTL+); and mice with tumors treated with 100 (SE100; *n* = 6), 200 (SE200; *n* = 4), or 400 mg/kg (SE400; *n* = 5) seed extract. The signal “-” indicates there were no animals with that score. Score intensity was minimal: +, moderate: ++, intense: +++.

**Table 5 cancers-15-02544-t005:** *Euterpe oleracea* seed extract effect on the lymphoid organs of Ehrlich solid tumor.

Treatment Group	Spleen Weight (g)	Spleen Cell Number (×10^4^)	Popliteal Lymph Node Weight (g)	Popliteal Lymph Node Cell Number (×10^4^)	Bone Marrow Cell Number (×10^4^)
Sham	0.00 ± 0.001	5601.7 ± 1452.8	0.17 ± 0.02	4.0 ± 6.15	616.3 ± 380.7
CTL−	0.05 ± 0.014 ^aaa^	10,538.3 ± 3309.3 ^a^	0.44 ± 0.06 ^aaa^	3381 ± 492.5 ^aaa^	321.3 ± 71.8
CTL+	0.0 ± 0.002 ^bb^	2603.3 ± 2507.2 ^bb^	0.19 ± 0.11 ^bb^	78 ± 25.1 ^bbb^	132.0 ± 59.5 ^aaa,b^
SE100	0.04 ± 0.017	6283.3 ± 2064.8 ^c^	0.30 ± 0.05 ^aa,ccc^	703 ± 505.9 ^a,b^	607.3 ± 397.9 ^cc^
SE200	0.04 ± 0.012	5775.8 ± 1229 ^b^	0.24 ± 0.037 ^aa,ccc^	887.5 ± 451.7 ^a,b^	205.3 ± 72.1
SE400	0.02 ± 0.007 ^b^	4818.0 ± 2407.2 ^bb^	0.20 ± 0.071 ^a,bb^	1000 ± 231 ^aa,b^	377.0 ± 205.1 ^c^

Groups treated with seed extract at 100 mg/kg (SE100), 200 mg/kg (SE200), and 400 mg/kg (SE400) were compared with the group without solid tumor (Sham) represented by “^a^”; the tumor-inoculated group treated with saline (CTL−) represented by “^b^”, and the tumor-inoculated group treated with cyclophosphamide at 20 mg/kg (CTL+) represented by “^c^”. Data represent the mean ± S.D. The statistical difference was represented by one letter for *p* < 0.05, two letters for *p* < 0.01, and three letters for *p* < 0.001 (ANOVA and Tukey´s post-hoc test).

**Table 6 cancers-15-02544-t006:** Histological scores in hepatic and renal tissues from all experimental groups.

Organ	Histological Features	Scores	Animal Groups					
			Sham	Ehrlich+				
				CTL−	CTL+	SE (mg/kg)		
						100	200	400
Liver	Mild diffuse vacuolar degeneration	0	6	-	-	-	-	-
		+	-	-	-	-	-	-
		++	-	-	4	3	1	2
		+++	-	6	2	3	3	3
	Moderate diffuse vacuolar degeneration	0	6	-	-	-	-	-
		+	-	2	-	-	-	-
		++	-	3	2	2	2	4
		+++	-	1	4	4	2	1
Kidney	Mild diffuse vascular congestion	0	6	-	-	-	-	-
		+	-	-	-	-	-	-
		++	-	2	5	4	2	5
		+++	-	4	1	2	2	-
	Moderate diffuse vascular congestion	0	6	-	-	-	-	-
		+	-	-	-	-	-	-
		++	-	3	2	3	-	1
		+++	-	3	4	3	4	4
	Mild multifocal vascular congestion	0	6	-	-	-	-	-
		+	-	3	-	1	-	-
		++	-	1	2	-	-	2
		+++	-	1	4	5	4	3

Mice without tumor induction (Sham; *n* = 6); mice with tumors treated with saline (CTL−; *n* = 6); mice with tumors treated with cyclophosphamide (CTL+); and mice with tumors treated with 100 (SE100; *n* = 6), 200 (SE200; *n* = 4), or 400 mg/kg (SE400; *n* = 5) seed extract. The signal “-” indicates there were no animals with that score. Score intensity was minimal: +, moderate: ++, intense: +++.

## Data Availability

The data are contained within the article.

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
