# Peer review of "Antitumor Effect of Açaí (Euterpe oleracea Mart.) Seed Extract in LNCaP Cells and in the Solid Ehrlich Carcinoma Model"

_cancers, 2023, doi:10.3390/cancers15092544_

Round 1

Reviewer 1 Report

In this study the authors investigated the effect of seed extract of açai against cancer cell in vitro and in vivo in animals with Erlich tumor model. The authors showed that açai seed extract possessed several secondary metabolites of which catechin/epicatechins are the dominants. They also indicated that the extract was active against LNCaP prostate cancer cells in vitro and exhibited promising in vivo efficacy against Erlich tumor model. Overall, the article reads well and the methods employed are well presented. My comments and suggestions are presented below.

11. Simple summary part needs to be revised. It contains a few grammatical errors and sentence flaws.  

22. Page 1, line 50-51: Palm and fruit extracts not exhibited in vitro antitumor activity, while fruit and seed extracts showed cytotoxic effects on LNCaP cells. How fruit extract can be active and inactive? Need to clarify this.

33. Why only 70% ethanolic extract was employed? It would have been excellent if you did solvent fractionation and activity-guided isolation and identification of potential compounds.

44. Why did you use docetaxel and cyclophosphamide as your positive controls for in vitro and in vivo studies, respectively? Why didn’t you use one of them in both studies?

55. Table 5: what do aa, bb, cc, aaa refer too? They should explicitly be explained under the table. In the same table for CTL+ the spleen weight was reported as 0.0±0.002. What does this mean? How can the average weight of the spleen be zero?

66. Apart from the ones presented, did you do any mechanistic investigations for your extract such as western blotting, apoptosis etc?

77. Have you determined the maximum tolerated dose of the extract in healthy female mice? If not why, and how did you proceed to in vivo study before you determine its maximum tolerated dose? How did you choose the three doses for the in vivo study?

87. Page 12, lines 398-400: “this result clearly demonstrates that at a dose of 400 mg/kg, acai seed extract was as effective as cyclophosphamide” in reducing tumor growth”. This statement is misleading that it doesn’t consider that cyclophosphamide was used at 20-fold lower dose than the extract. Hence, tone-down your claim of effectiveness and try to maximize your finding that the extract even at the highest dose used showed better safety profile than cyclophosphamide.

99. Acute toxicity is evaluated following single limit-dose administration to healthy, female mice. Usually, 2 gm/kg of an extract is administered and mice observed for up to 14 days for signs and symptoms of toxicity. However, the way the authors claimed about absence of acute toxicity in this article is misleading. The study was conducted with male mice having tumor and dose was administered repeatedly for 15 days, and such study can’t be considered as acute toxicity study. Hence, the authors should change their claim about acute toxicity outcomes.    

Author Response

  1. Simple summary part needs to be revised. It contains a few grammatical errors and sentence flaws.

ANSWER: The summary was revised and alterations were made to correct grammatical errors and sentence flaws.

  1. Page 1, line 50-51: Palm and fruit extracts not exhibited in vitro antitumor activity, while fruit and seed extracts showed cytotoxic effects on LNCaP cells. How fruit extract can be active and inactive? Need to clarify this.

ANSWER: Thank you for your comment. The correct sentence is “Palm and pulp extracts not exhibited …”. It was adjusted in the manuscript (Line 50).

  1. Why only 70% ethanolic extract was employed? It would have been excellent if you did solvent fractionation and activity-guided isolation and identification of potential compounds.

ANSWER: Thank you for your comment. It is true that the use of extract fractions would help in identifying possible components with the properties of interest. However, that was not our goal. In this work, we focused on total extracts from different parts of the plant to identify possible regions with greater or lesser biological potential. This strategy aims to allow the use of all parts of this plant species. Thus, valuing its cultivation and conservation. We do not rule out the possibility of refining the extracts, but this would be a future work.

  1. Why did you use docetaxel and cyclophosphamide as your positive controls for in vitro and in vivo studies, respectively? Why didn’t you use one of them in both studies?

ANSWER: Thank you for your comment. We choose the reference drugs according to the current clinical use regarding each cancer type, as well as its use in experimental models. Docetaxel is a well know therapeutic option in the treatment of prostate cancer (doi: 10.20471/acc.2022.61.s3.5), and thus was used as reference drug in assays against prostate cancer cell line in vitro. In addition, its direct effect on tubulin (no need for metabolic conversion) allows its use in vitro assays (doi.org/10.1016/j.jddst.2020.101959). Cyclophosphamide is a drug that reduces the risk of recurrence and mortality in early breast cancer (Ejlertsen, 2016), and modulates inflammation and immune response in Ehrlich model, being used as reference drug in in vivo breast cancer experimental models (doi: 10.1016/j.lfs.2005.10.006; doi: 10.1093/jnci/58.6.1759). Moreover, the use of cyclophosphamide in vitro would be unfeasible, since its action depends on a primary conversion that occurs in the liver (doi:10.1038/nrclinonc.2009.146). We added this clarification in the manuscript (line 506-509; 560-564).

  1. Table 5: what do aa, bb, cc, aaa refer too? They should explicitly be explained under the table. In the same table for CTL+ the spleen weight was reported as 0.0±0.002. What does this mean? How can the average weight of the spleen be zero?

ANSWER: Thank you for your comment. Each letter represents comparisons with specific groups. “a” was used when the statistical comparison was made in relation to the Sham group, “b” in relation to the CTL- group and “c” in relation to the CTL+ group. Regarding the number of letters, they are a way of representing the degree of statistical difference found in relation to the p value. Thus, statistically difference was represented by one letter when value p <0.05, two letters for p <0.01, and three letters for p <0.001. This information was already placed in the figure caption.

  1. Apart from the ones presented, did you do any mechanistic investigations for your extract such as western blotting, apoptosis etc?

ANSWER: All the analysis performed until now are presented in the manuscript. We are looking for financial support to carry out the other suggested experiments regarding mechanistic investigations.

  1. Have you determined the maximum tolerated dose of the extract in healthy female mice? If not why, and how did you proceed to in vivo study before you determine its maximum tolerated dose? How did you choose the three doses for the in vivo study?

ANSWER: Thank you for your comment. The maximum tolerated dose was not determined in this study. We choose the doses according to the literature reports and studies of our research groups. In literature reports, there are studies using 300 mg/kg/day of Açaí seed extract (ASE) rich in proanthocyanidins during 12 weeks (doi: 10.1016/j.cbi.2021.109721), and our group already used a single treatment dose of 1,000 mg/kg with catechin-rich Açaí Seed Extract  similar to that used in this research (doi: 10.3390/foods10051014). In addition, catechin, the major compound in açaí seed extract, has also toxicity data that helped to set the treatment dose of this study. In PubChem, the LD50 of catechin administered by oral route is reported as >10 g/kg in mouse. An in silico analysis using pkCSM performed by us predicted no hepatotoxicity, and Oral Rat Chronic Toxicity (LOAEL) of 316.2 mg/kg bw/day of catechin, that considering the catechin concentration of 45% in our extract, would allow the use of dose up to 702.6 mg/kg bw/day of açaí seed extract. All these data were taken into account, and were endorsed by the absence of histological hepatic and renal tissues changes related to the administration of E. oleracea seed extract, that in fact provided an weight gain. We added this information in the discussion section (line596-605). Regarding the sex of the animals, we did not use females because the Ehrlich tumor is a murine mammary carcinoma, which can suffer hormonal influence (doi.org/10.3181/00379727-116-29475). Therefore, the use of males removes the bias of the estrous cycle, diminished the influence of hormonal fluctuation.

  1. Page 12, lines 398-400: “this result clearly demonstrates that at a dose of 400 mg/kg, acai seed extract was as effective as cyclophosphamide” in reducing tumor growth”. This statement is misleading that it doesn’t consider that cyclophosphamide was used at 20-fold lower dose than the extract. Hence, tone-down your claim of effectiveness and try to maximize your finding that the extract even at the highest dose used showed better safety profile than cyclophosphamide.

ANSWER: Thank you for your comment. It is true that the dose of the extract used is 20 times higher than the chemotherapy dose. However, it should be taken into account that it is a crude extract, so its effects will be less than an isolated and synthesized drug (doi.org/10.3390/medicines2030251). We tried to improve our statement by changing the sentence to: this result clearly demonstrates that at a dose of 400 mg/kg, acai seed extract was as effective as the dose of cyclophosphamide used (20 mg/kg). (line 399-401)

  1. Acute toxicity is evaluated following single limit-dose administration to healthy, female mice. Usually, 2 gm/kg of an extract is administered and mice observed for up to 14 days for signs and symptoms of toxicity. However, the way the authors claimed about absence of acute toxicity in this article is misleading. The study was conducted with male mice having tumor and dose was administered repeatedly for 15 days, and such study can’t be considered as acute toxicity study. Hence, the authors should change their claim about acute toxicity outcomes.

ANSWER: Thank you for your comment. What we affirmed was that at these doses of extract, in animals carrying the tumor, no signs of toxic effects associated with the treatment were detected (as demonstrated by liver and kidney histology). It is important to note that we use the term acute since we consider a chronic treatment when it is longer than 90 days, since this specific time is considered sub-chronic (OECD guideline 452; OECD guideline 408). In addition to the OECD guides, several other works use terms similar to the ones we used in this work (doi.org/10.1016/j.fct.2016.09.018; doi.org/10.1016/j.chemosphere.2017.09.025). Thus, our use of the term acute is more related to the duration of treatment than the number of doses. Nevertheless, we removed the term “acute” from the manuscript to clarify this point.

Reviewer 2 Report

In this manuscript, the authors evaluated the antitumor features of E. oleracea in vitro and in vivo via LNCaP prostate cancer cell line and solid Ehrlich tumor in mouse model, respectively. Catechin and epicatechin were characterized from the extraction of seed. IC50 was detected in LNCaP, and the extracts show selective cytotoxicity to tumor cells other than normal cells. Some tumor inhibition and immunoprotecting properties have been found in the mice treated by 400 mg/kg extracts oral treatment. However, there are some concerns need to be addressed.

1. The author identified most of the components in the extracts. However, in this manuscript, they haven’t figured out which component is the most potent one to have the tumor inhibition and immunoprotecting properties among all the components in the extracts. Although catechin and epicatechin have been shown large amount in the extracts, in the later experiments, none of pure catechin and epicatechin were involved into the experiments as control to compare with the extracts. Therefore, I would suggest use pure catechin and epicatechin in the cytotoxicity studies or mice experiments to compare with the extract.

2. The author claimed that seed extract for LNCaP cells inhibition is not time dependent. However, the shortest incubation time is 24 h which may be long enough for the extract to kill the tumor cells. I would suggest use 2h, 4h, 8h, and 12 h incubation time with the cells to confirm if the result is time dependent or not.

3. There is lack of untreated control group of the cell viability in fig 2C, please add the data of untreated control group.

4. Please uniform the background of fig 2. In figure 2 D,E,F,and G show gray background while others show white.

5. Please correct ‘ACU’ on Y axis of fig 4 C to AUC.

6. Please add scale bar and symbol to point out necrosis or inflammation infiltration and others in Fig. 5.

7. Please center the Table 5.

8. Please explain why the cyclophosphamide has been chosen as positive control to treat Ehrlich solid tumor in mice. The cyclophosphamide showed significant immunosuppression properties. In addition, the cytokine production level of TNF, IL-6 and IFN-γ are too low to claim if there is an inflammation has been triggered in the solid tumor mouse model. The meaning of ‘immunoprotection’ is also unclear. Please confirm the production level of this cytokines to confirm the inflammation status and clearly explain what does the ‘immunoprotection’ mean.

Author Response

  1. The author identified most of the components in the extracts. However, in this manuscript, they haven’t figured out which component is the most potent one to have the tumor inhibition and immunoprotecting properties among all the components in the extracts. Although catechin and epicatechin have been shown large amount in the extracts, in the later experiments, none of pure catechin and epicatechin were involved into the experiments as control to compare with the extracts. Therefore, I would suggest use pure catechin and epicatechin in the cytotoxicity studies or mice experiments to compare with the extract.

ANSWER: Thank you for your comment. It is true that we did not test the major constituents of the extracts to assess their antitumor effect, as this was not our objective. The present work aims to test the crude extract of different parts of the plant in order to find possible regions with greater or lesser biological properties. This will make it possible to know (and value in a certain way) this plant species. Furthermore, the use of an extract is closer to our reality of use than isolated substances. The study of isolated substances would only take place in a later work, whose production would depend directly on the publication of the present work.

  1. The author claimed that seed extract for LNCaP cells inhibition is not time dependent. However, the shortest incubation time is 24 h which may be long enough for the extract to kill the tumor cells. I would suggest use 2h, 4h, 8h, and 12 h incubation time with the cells to confirm if the result is time dependent or not.

ANSWER: Thank you for your comment. We designed our experiments thinking of a medium to long-term in vivo treatment. We agreed with you that shorter treatment times would change our statement related to the lack of time-dependent activity observed in some treatment scenarios and we made changes in the manuscript to clarify this point (line 353). In addition, we aim to carry out mechanistic studies and certainly shorter times such as suggested for you will be evaluated in a later work.

  1. There is lack of untreated control group of the cell viability in fig 2C, please add the data of untreated control group.

ANSWER: The untreated control group was used as reference group to calculate each percentage of survival, and thus it is considered always as 100% ± SD, and it is represented in concentration 0 ug/mL of X axis. The treatment at 0 ug/mL was added to all the graphs of viability results.

  1. Please uniform the background of fig 2. In figure 2 D,E,F,and G show gray background while others show white.

ANSWER: The gray background was removed.

  1. Please correct ‘ACU’ on Y axis of fig 4 C to AUC.

ANSWER: Thank you for the observation, it was corrected in the figure.

  1. Please add scale bar and symbol to point out necrosis or inflammation infiltration and others in Fig. 5.

ANSWER: We added scale bars, arrows and lettering as requested.

  1. Please center the Table 5.

ANSWER: Thank you for your comment. For same reason the journal system altered the table position. We corrected the position of the Table 5.

  1. Please explain why the cyclophosphamide has been chosen as positive control to treat Ehrlich solid tumor in mice. The cyclophosphamide showed significant immunosuppression properties. In addition, the cytokine production level of TNF, IL-6 and IFN-γ are too low to claim if there is an inflammation has been triggered in the solid tumor mouse model. The meaning of ‘immunoprotection’ is also unclear. Please confirm the production level of this cytokines to confirm the inflammation status and clearly explain what does the ‘immunoprotection’ mean.

ANSWER: Thank you for your comment. Cyclophosphamide is effective in controlling the growth of Ehrlich's solid tumor, according to the literature and previous work by the group. Thus, this drug was used in the in vivo model (doi.org/10.3390/md18110541; doi.org/10.1186/s12906-021-03305-2). The immunosuppressive effects of cyclophosphamide are well known, but they are overlapped by the antineoplastic effect of the drug (doi:10.1038/nrclinonc.2009.146). Furthermore, many studies believe that part of the drug's antitumor effect is due to its effect on the immune system (doi.org/10.1016/j.ctrv.2015.11.005).

Regarding the inflammatory issue, the inflammatory nature of the Ehrlich's tumor is very well established (doi.org/10.1016/j.vascn.2014.09.001). Despite this, the solid form of the tumor tends to have a more discreet systemic inflammatory profile, when compared with the ascitic form (doi.org/10.1016/j.lfs.2020.118578). In our model, we can clearly observe the inflammatory effect of the tumor by the increased serum concentrations of TNF, MCP-1 and IL-6. Which, despite not being at high levels, have a statistically significant difference between the Sham and the CTL- group (Figure 7). On top of that, if we take into account the percentage increase based on the sham group, all cytokines had at least two-fold greater production in the group with tumor.

Finally, the term immunoprotection was used as a way of demonstrating that the extract had an antitumor action without causing major deleterious effects on the immune system. Side effects that were found in the group treated with cyclophosphamide, for example. This statement can be evidenced by the similarity of cellularity of the lymphoid organs between the sham groups and those treated with the extract (Table 5). In addition, it is important to highlight how the extract was able to modulate the immune response by decreasing IL-6 and increasing IFN (at a dose of 400 mg/kg), two important events for tumor control. Thus, it is possible to say that the extract had an antitumor effect without major side effects on the immune system (such as those found in the cyclophosphamide group), hence the use of the expression immunoprotection.

Round 2

Reviewer 2 Report

The revised version addressed all of my concerns.